# Metabolic Mechanisms and Potential Therapeutic Targets for Prevention of Ovarian Aging: Data from Up-to-Date Experimental Studies

**DOI:** 10.3390/ijms24129828

**Published:** 2023-06-06

**Authors:** Konstantinos Valtetsiotis, Georgios Valsamakis, Evangelia Charmandari, Nikolaos F. Vlahos

**Affiliations:** Second Department of Obstetrics and Gynaecology, Aretaieion University Hospital, National and Kapodistrian University of Athens Medical School, 115 28 Athens, Greece; k.valtetsiotis@gmail.com (K.V.); evangelia.charmandari@googlemail.com (E.C.); nfvlahos@gmail.com (N.F.V.)

**Keywords:** longevity, aging, oocytes, reproduction, fertility

## Abstract

Female infertility and reproduction is an ongoing and rising healthcare issue, resulting in delaying the decision to start a family. Therefore, in this review, we examine potential novel metabolic mechanisms involved in ovarian aging according to recent data and how these mechanisms may be addressed through new potential medical treatments. We examine novel medical treatments currently available based mostly on experimental stem cell procedures as well as caloric restriction (CR), hyperbaric oxygen treatment and mitochondrial transfer. Understanding the connection between metabolic and reproductive pathways has the potential to offer a significant scientific breakthrough in preventing ovarian aging and prolonging female fertility. Overall, the field of ovarian aging is an emerging field that may expand the female fertility window and perhaps even reduce the need for artificial reproductive techniques.

## 1. Introduction

Infertility is a global problem on the rise. The WHO defines it as a condition of the reproductive system that can be diagnosed when there is a “failure to achieve a clinical pregnancy after 12 months or more of regular unprotected sexual intercourse” [1]. It occurs due to four broad causes: lifestyle choices, inheritable factors, health conditions and aging [2], with a degree of overlap between each of these factors. A worldwide systematic study found that in 2010, 48.5 million couples of reproductive age suffered from infertility [3], with the absolute value having increased since then due to population growth. Interestingly, this is possibly a result of the decision to have children at a later age, as opposed to a relative decrease in fertility [3].

In Western countries, women’s first-time pregnancies occur 3–5 years later than 50 years ago [4]. For the first time since demographic data started being recorded in the USA, the 30–34 age group had a higher birth rate compared to the 25–29 group [5]. This has come as a result of dropping birth rates in the younger group and rising birth rates in the older group. The EU displays a similar trend; in Ireland, Spain, Italy, the Netherlands, Greece and Cyprus, women have their first birth when they are aged 30 or older, with many other mothers on the brink of reaching this age group [6]. These data pertain to the necessity of maintaining fertility in the aging population. Above all, nowadays, women have a shorter remaining reproductive period due to their priorities, making their fertility a greater societal concern.

The primary indicators of fertility in women are the quantity and quality of oocytes. The current well-established known factors and mechanisms of ovarian aging have already been described [7,8]. These include DNA damage and problems during meiosis such as spindle formation, mis-segregation due to recombination too close to the centromere and chromosomal cohesion [8]. Oxidative damage caused by reactive oxygen species (ROS) accounts for a large part of the DNA-related damage. There is also age-related downregulation of transcription for genes involved in antioxidative pathways, which leads to oxidative damage and subsequent apoptosis [7]. In recent years, however, other more specific metabolic mechanisms that influence ovarian aging have been discovered. These are connected to at least one of the mechanisms listed above, and sometimes perhaps more. Though their mechanism may not be clearly understood, studies have examined demonstrate a positive correlation with inhibiting ovarian aging when they are expressed/upregulated, and a negative correlation when they are not expressed/downregulated. These include the PI3K/AKT/mTOR pathway, sirtuins and circulating miRNA. The PI3K/AKT/mTOR pathway, when inhibited, appears to protect against follicular atresia. At the same time, when promoted, the pathway promotes folliculogenesis and accelerates the depletion of the follicular pool [9]. Sirtuins are involved in various protective mechanisms such as the removal of oxidative agents and genetic mutators [10]. Finally, miRNAs appear to be regulators of aging and apoptotic mechanisms [11]. The known pathways for ovarian aging are, therefore, being expanded, and any new mechanisms may aid in discovering novel targets for ovarian aging prevention. These new mechanisms are particularly important as they may provide avenues for creating regiments that could delay/inhibit ovarian aging, instead of medical research focusing solely on its treatment through IVF.

Of the well-established mechanisms, mitochondrial DNA is more prone to mutations than nuclear DNA due to less effective repair mechanisms [12]. In mitochondria, similar downregulation occurs, leading to mitochondrial DNA (mtDNA) damage, making it more unstable and subsequently decreasing oocyte quality [7]. Another factor is chromosomal cohesion; the sister chromosomes need to stay attached and subsequently detach at specific intervals throughout meiosis, and failure to do so can result in aneuploidy. The mechanism that prevents premature degradation naturally degrades with age [8]. In the granulosa cells, mtDNA damage leads to apoptosis. The decrease in granulosa cells changes the follicle’s ratio of androgens to oestrogens due to a decrease in aromatase levels [7,12]. This, in turn, creates problems in postfertilisation maturation, thus decreasing the quality of the follicles. Another factor is the ovarian microenvironment. Aging increases the frequency of chronic low-grade inflammation, which, over time, can decrease the quantity and quality of oocytes [8]. Lastly, telomere length is associated with oocyte quality and may also be associated with the age at which menopause is reached [8]. This may be due to a general decrease in telomere length amongst all oocytes, more oocytes reaching a critical minimum threshold [13] or other factors [14]. The above-mentioned processes are only understood to be factors; the exact mechanisms of both quantitative and qualitative degradation remain unknown and, as such, we intend to further explore the methods and mechanisms involved, linking them with potential treatments based on current data.

There are limited studies on the medical treatments that could influence ovarian aging: metformin, DHEA, stem cells, stem cell derivatives, caloric restriction, caloric restriction mimetics, mitochondrial transfer, hyperbaric oxygen treatment, macrophage-derived extracellular vesicles, cell-free fat extract, and TrKB agonists. Stem cell research in particular is quite promising as treatments used in primary ovarian insufficiency (POI) have displayed longevity-related effects in healthy mice that could potentially be targeted to the ovaries for fertility preservation [15,16,17]. Such experimental treatments present a potential approach for someday creating regiments to delay and extend ovarian aging.

Our aim is to explore the recent literature for new promising molecular/metabolic mechanisms and treatments that could possibly prevent and delay ovarian aging and, thus, expand the reproductive timespan.

## 2. Metabolic and Molecular Mechanisms of Ovarian Aging

The number of oocytes present in females varies throughout life. During the first 5 months before birth, the numbers increase steadily to about 7 million. By the time of birth, the oocytes decrease to around 1 million through atresia [18,19]. Atresia is the natural process through which ovarian function is regulated. For a long time, follicular atresia has been hypothesised to be the primary cause of menopause and, thus, infertility [20]. Atresia works by activating the apoptotic pathways [21], leading to only around 400 oocytes maturing to the degree necessary for ovulation and fertilisation [21,22]. Working counter to atresia is the process of folliculogenesis; the mechanism that develops, matures and selectively releases oocytes to the uterus [23]. The balance of atresia to folliculogenesis is a complex and tightly controlled system. Atretogenic factors include tumour necrosis factor-α (TNF-α), Interleukin-6 and androgens [22]. Androgens, however, are also important for the normal functioning of oocytes in natural physiological concentrations. The PI3K pathway has two ovarian functions. First, when upregulated, it triggers the maturation of primordial follicles into primary follicles. Second, in low doses, it prevents atresia from occurring in the primordial follicle pool in its dormant state. Testosterone promotes the PI3K pathway in follicles, thus ensuring both the above crucial physiological functions can be maintained [24]. Androgens play an important role in stimulating atresia in further follicular stages to ensure mono-follicular activation in humans [25]. Finally, their production in theca cells also leads to the production of oestradiol, the primary female sex hormone, in granulosa cells [26]. There have been some attempts to use androgens for fertility treatment, which we explore in Section 4.4.1 but any attempts to disrupt the delicate balance that androgens have in the female reproductive system should be met with caution.

Folliculogenic factors include gonadotropins and downward intermediates such as Interleukin-1β, insulin-like growth factor-1 and oestrogens [22]. As age increases, besides the quantitative loss of oocytes, the risk of aneuploidy increases. Aneuploidy in oocytes is the unequal separation of chromosomes during meiosis I or II, resulting in gametes with less/more than a single set of 23 chromosomes [27].

## 3. General Factors Involved in Ovarian Aging

Ovarian aging mechanisms are still unclear, as is their effect on oocyte quality/quantity. It would, however, be beneficial to examine the established and novel mechanisms so that we may then theorize on how they may be targeted.

The rate at which atresia occurs is largely accelerated as a woman approaches menopause when compared to the more linear decline seen before the mid to late thirties [20,22]. For a long time, the modelling for ovarian follicular loss was thought to best fit a biphasic decrease [22] (Figure 1). This is because the rate of atresia seems to increase twofold in the late 30s (37–38) when a threshold of around 25,000 follicles is exceeded. If this modelling was correct, without the twofold increase, follicular depletion would not occur until around 70 years of age [22]. More recent evidence, however, suggests that the follicular decay more closely fits a power model of decay [28], suggesting a more gradual exponential decrease that may not occur at exactly 37 years old and without there being a particular follicle threshold that triggers accelerated recruitment.

### 3.1. Genetic Factors Involved in Ovarian Aging

The rate of atresia may be directly linked to the quality of the oocytes instead of the number of follicles. In mice that lack CHTF18, a protein that ensures chromatid cohesion and stable DNA replication, the rate of follicular atresia is notably higher than in CHTF18+ mice [29]. The increase in atresia also comes with an increase in aneuploidy and DNA double-strand breaks (DSB). Along the same lines, women with BRCA1 mutations experience earlier menopause along with higher aneuploidy [30,31]. This pattern is also seen in BRCA2 mutations [31], with primordial follicular density demonstrating a significant decrease compared to controls, especially after 30 years of age. The age of menopause is also linked to other genes related to homologous recombination such as BRE, MCM8 and INO80 [32].

It has been hypothesised that oocytes collect double-strand breaks in their DNA during their lifetime [33]. Titus et al. [33] showed that several genes involved in ATM-mediated DNA DSB repair pathways (BRCA1, ATM, MRE11, etc.) showed a decrease in expression after 37 years of age. However, the data were explained using a biexponential model of follicular decay [22] and, as discussed above, the decrease could be more gradual than the authors believe. The results could still fit into the more gradual power model (Figure 1). We have seen above [30,31] that genetic BRCA1/2 mutations do lead to higher follicular atresia and a younger menopausal age. Taken together, the natural rate of atresia, which leads to menopause at around 45 years of age, may have the potential to be targeted for preventative treatment. If such treatment were successful, it would prolong menopause beyond this age limit and, thus, extend the female fertility window.

### 3.2. DNA Damage

DNA damage in the oocytes is accumulated over the reproductive lifespan. It may be caused by intrinsic factors, such as accrued replicated stress, ROS and hydrolysis, or by extrinsic factors, such as genotoxic substances, mutagens and radiation [34]. All these factors provide various mechanisms for damage incidents to occur on the oocyte DNA, whether structural or intrinsic. These can be thought of as “hits”, and when a certain number of hits is reached, the oocyte will be unsuitable for development into a foetus. It will then be either eliminated through apoptotic atresia or cause aneuploidy. The “hits” hypothesis [35] suggests that even if one mechanism for DNA damage is stopped or prevented, others will accumulate enough to still cause infertility. As they accumulate, they may have a domino effect, leading to exponential oocyte loss; the malfunction of one repair mechanism may have a detrimental effect on others, causing a cascade that tips over the delicate atresia/folliculogenesis equilibrium.

### 3.3. Cohesins

Cohesins are another factor leading to aging when they fail. They are the proteins that hold the sister chromatids together and are not renewed once formed; the same cohesins from foetal development must provide structural DNA support for up to 4 decades later [36,37]. After all, cohesins in oocytes sustain the sister chromatids in prophase I until the oocyte is selected for ovulation [37]. The fact that there is no cohesin turnover, combined with inexistent repair mechanisms, gives rise to the theory that cohesins play a vital role in causing aneuploidy [37], as sister chromatids separate prematurely. The “cohesin deterioration” hypothesis [38,39,40] suggests cohesion deterioration is the leading cause of aneuploidy, though the specific mechanism remains unclear [39]. It may be due to cohesin deterioration, primarily around the centromere [40]. Although cohesins also attach to specific points on the chromosomal arms [41], centromere cohesion appears to critically deteriorate first, with the mechanism for this deterioration remaining unclear [40].

### 3.4. Telomere Length

Telomere length may also influence oocyte quality. In a study with 120 POI patients, Xu et al. showed shorter telomere length in the granulosa cells compared to control [42]. Miranda-Furtando et al. [43], on a similar note, showed similar results in a study with 127 POI patients. Similarly, a significant difference in telomere length compared to control has been observed in patients with occult POI [44]. From these findings, it may be suggested that a short telomere length (and, therefore, reduced telomerase activity) may trigger atresia once a critical length has been reached. Additionally, though the cause of some forms of POI is still unknown, telomere length should be capable of acting as a predictive tool for the age of menopause and may furthermore be targeted as a mechanism to extend ovarian health for a longer time.

### 3.5. Oxidative Damage

A final factor is oxidative damage through deactivated antioxidative pathways. Ovarian aging has been shown to be linked to a downregulation of the pathways that prevent ROS from causing oxidative damage in both mitochondrial and nuclear DNA. On a non-human primate study, Wang et al. [45] demonstrated that a downregulation of transcription for key antioxidant mechanisms occurs with aging. This was shown by the statistically significant increase in DNA oxidation biomarker 8-OHdG and DNA damage biomarker γH2AX, among others. Such a study is important given that direct human ovarian samples are difficult to obtain, and it can provide a close model for ovarian aging. Oxidative damage may play a wider role and may be the predominant factor affecting all the mechanisms discussed above; it has been demonstrated to be linked to telomere shortening, genetic instability and subsequent apoptosis [46] through ovarian dysfunction (Figure 2). Furthermore, oxidative damage is shown to affect both mtDNA and DNA repair mechanisms [47]. In conclusion, the prevention of oxidative damage may be the key factor in preventing ovarian aging through the previously discussed pathways.

mtDNA is especially prone to DNA damage as it relies on the cell’s endogenous antioxidant mechanism rather than a mtDNA-specific mechanism for damage repair [48]. Mitochondria are in greater demand in oocytes compared to somatic cells, given the considerable ATP required for fertilisation and blastocyst formation. As such, mature (metaphase II) oocytes have up to 100,000 mitochondria [48], compared to the few thousand that may be found in somatic cells [49].

## 4. Potential Molecular/Metabolic Treatments for Prevention of Ovarian Aging

### 4.1. Pharmaceuticals

Four prescription pharmaceuticals were identified with regard to delaying ovarian aging (Table 1).

#### 4.1.1. Metformin

Metformin is an antidiabetic drug used primarily for type 2 diabetes mellitus. It has some indications of having antiaging effects, although the results are contested and controversial [54]. Qin et al. conducted a mouse study looking at metformin supplementation and its effects on ovarian aging [50]. The mice were given 100 mg/kg metformin for 6 months, with a concurrent control group. The metformin group had significantly higher primordial and primary follicle counts. There was no significant difference between other follicle types. Moreover, the metformin group had significantly lower 8-OHdG, a DNA oxidative damage biomarker, as well as 4-HNE, a lipid damage biomarker. P16, a senescence and aging biomarker [55], was also significantly lower. The researchers had hypothesised that SIRT1 levels would be stimulated, which, as we have seen earlier, should inhibit ovarian aging by decreasing oxidative damage and ROS production. As predicted, SIRT1 levels were significantly higher in the metformin group compared to the control group. This study concluded that although no anti-apoptosis effects were noted, metformin can inhibit ovarian aging through its SIRT1-stimulating effects. This study has the advantage of having performed the experiment for 6 months, whereas most studies opt for a shorter timeframe. Therefore, we could expect these findings to be a good predictor of long-term metformin use, and further studies should be encouraged.

#### 4.1.2. Visfatin

Visfatin is an adipocytokine produced by visceral adipose tissue [56] with insulin-mimetic actions. A study investigated its effects on mice [51]. The researchers hypothesised that since the activation of dormant primordial follicles can aid in fertility treatment in cases of POS [57], triggering the underlying mechanism of activation may aid in restoring fertility. Visfatin is known to trigger the mTOR/PI3K signalling pathways [58,59] and may, therefore, trigger the mechanism to restore fertility. In the study, 18-month-old female mice (human equivalent 60–70 years old) were injected with 0 (control), 500 or 1000 ng/mL Visfatin thrice over a period of 6 days. The two Visfatin groups had a significantly higher number of primary, secondary and early antral follicles. Additionally, both groups had a significantly lower atretic follicle count compared to the control. In a second experiment of the same study, mice split into the same groups were allowed to mate instead of having their ovaries harvested. They were first injected with 5 IU of equine and human chorionic gonadotropin to trigger superovulation. The Visfatin groups both had a significantly higher number of zygotes retrieved. The 500 ng/mL group had a significantly higher number of pregnancies compared to the control, as the control had no pregnancies. While the 1000 ng/mL group outperformed the control in terms of pregnancies, there was no statistical difference. The efficacy of 500 ng/mL is supported through the relative mRNA levels of the mTOR/PI3K signalling pathway components tested (4EBP1, S6K1, RPS6). Two of the three tested components (S61K, RPS6) had a significant increase in the 500 ng/mL group compared to the control, while the 1000 ng/mL group did not have any significant difference. This showed that the mTOR/PI3K pathway was indeed activated at the lower dose (as hypothesised), explaining the higher number of pregnancies and offspring. Therefore, Visfatin may be a promising treatment option as an artificial reproductive option to perhaps allow post-menopausal women to conceive.

#### 4.1.3. Sphingosine 1-Phosphate

Sphingosine 1-phosphate, S1P, is a naturally occurring sphingolipid. Its ratio when compared to ceramide determines if a cell will undergo ceramide-induced cell death [60]. As such, two studies hypothesised that its supplementation could decrease the activation of this pathway and decrease the likelihood of spontaneous apoptosis in ovarian follicles [52,53]. Guzel et al. [52] performed an in vitro study on human cortical tissue samples from young (31-year-old mean age) human patients with ovarian endometrioma. The samples were split into groups and incubated with S1P concentrations of either 0 (control), 200 or 400 μM for 4 days. Both concentrations had significantly higher primordial and secondary follicle counts compared to the control. Both also had significantly lower cleaved-caspase 3 compared to the control, suggesting lower apoptosis.

Although the 400 μM concentration outperformed 200 in all three counts, the change was not significant. The researchers concluded that S1P inhibits spontaneous apoptosis in vitro. However, they warned that as the samples were all from ovarian endometrioma patients, it may be that they cannot be extrapolated to the general population. Nonetheless, the study shows at least the potential in inhibiting the ceramide-induced pathway. Mumusoglu et al. [53]. performed an in vivo rat study testing the effect of S1P administration in rats. The rats were 10 months old (~30 years old in human years) and administered either 0, 0.1 mg/kg or 1 mg/kg Fingolimod, a S1P analogue, for 60 days. The study found that the 1 mg/kg group had significantly higher AMH mean values compared to the control. The 0.1 mg/kg group had higher AMH values as well, but these were non-significant. Both groups also had a higher-non apoptotic follicle ratio compared to the control. These in vivo results support the cleaved caspase-3 measurements of Guzel et al. [52] in that S1P supplementation decreases spontaneous apoptosis and, therefore, preserves the follicular reserve. Altogether, S1P is a good pharmaceutical candidate for ovarian aging prevention given its follicular apoptosis-inhibiting properties.

### 4.2. Stem Cells

Studies regarding stem cells found them to be promising in terms of their potential to delay ovarian aging (Table 2).

#### 4.2.1. Human Amniotic Mesenchymal Stem Cells (hAMSC)

In one study, human amniotic mesenchymal stem cells were injected into mice [17]. The mice were transplanted with hAMSC and subsequently euthanised for examination over a period of 4 weeks. The hAMSC group of mice had significantly higher primordial, primary, secondary and antral follicle counts compared to the control. On a similar note, the AMH levels over the four weeks rose to the levels of the young control group, a significant difference compared to that of its age-matched control, which, as expected, was much lower.

As Ding et al. [17] had established a positive effect of hAMSC transplantation, they set forth to examine its causes. hAMSC lines were found to have a very high increase in EGF and HGF levels (epidermal growth factor and hepatocyte growth factor, respectively; EGF promotes oocyte maturation [68] and HGF promotes cell proliferation and inhibits apoptosis [69,70]) compared to a control T-cell line and this was hypothesised to be one of the primary reasons for the positive effects seen. Indeed, direct injection of EGF and HGF combined in mice was found to significantly increase total follicle count. Therefore, the researchers concluded that hAMSCs inhibit ovarian aging indirectly through their secretion of EGF and HGF. Through these results, we have seen that both hAMSC transplantation and EGF and HGF injections have the potential to delay ovarian aging.

#### 4.2.2. Umbilical Cord Mesenchymal Stem Cells (UC-MSC)

Mesenchymal stem cells can also be harvested from the umbilical cord. They have various advantages such as being less ethically controversial compared to other sources and having low immunogenicity [71].

A recent study combined umbilical cord mesenchymal stem cells with autocrosslinked hyaluronic acid (HA). The reasoning behind the HA addition is that it has been shown to act as an effective cell “scaffold” in other cell types [72] and, as such, it may have a similar effect on ovarian cells. First, the HA was tested to see if the retention of UC-MSCs was prolonged with its use, and its usage at 0.3 mg/mL significantly increased cell viability in vitro compared to the control. The second part of the experiment was seeing if UC-MSCs with HA could rescue follicular loss induced by VCD, which, as we have seen, is a commonly used ovotoxic research chemical. The primary follicle and total follicle count were indeed significantly higher in the UC-MSC group compared to age-matched VCD controls. However, primordial, secondary and antral follicles had no significant difference, so we should be cautious in inferring too much from the VCD results alone. Then, they determined that a paracrine mechanism was at play. The paracrine mechanism was found to be associated with the PI3K-AKT pathway. It was then identified that HGF was its key component, which we have already examined in Section 4.2.1 and has traits that inhibit ovarian aging. Most importantly, when administered in aging mice, the UC-MSCs significantly increased primary, secondary, antral and total follicle counts. During a 6-month period of mating after injection, the mean number of pregnancies per mouse was the same between the UC-MSC group and age-matched controls. However, per pregnancy, the litter size was significantly larger in the UC-MSC group compared to age-matched controls. Therefore, UC-MSC transplantation in the ovaries could have value as a treatment of ovarian aging.

Another study looking at UC-MSC combined its application with autocrosslinked hyaluronic acid (HA). Jiao et al. [62] transplanted 9-month-old mice (aged) with 5 μL UC-MSC mixed with HA 0.3 mg/mL. First, they established that UC-MSC combined with HA had a significant effect on UC-MSC properties. UC-MSCs with both had significantly higher levels of hepatocyte growth factor, vascular endothelial growth factor and stem cell factors. After the experiment duration, when compared to the age-matched controls, the mice with UC-MSC and HA had a significantly higher number of primary, secondary, antral and total follicles. The mice that were allowed to mate also had a significantly higher number of pups per litter. This provides further evidence of the potential of UC-MSC treatment, while also suggesting that its usage may be modified for even better efficacy, as may be seen in the following study too.

Instead of using the stem cells themselves, one could use the exosomes they produce. These are thought to be the primary method through which MSCs exert their effects, as they have repairing and regenerative properties [73]. These extracellular vesicles can interact with many cell types and can restore homogenesis in the tissue microenvironment they are in [73], including in oocytes. One experiment looked at their effects on mice. Ovaries from new-born mice were incubated and administered UC-MSC-exos (Figure 3), which increased mTOR/PI3K pathway activation. Afterwards, older, 10-month-old mice (~50 in human years) were injected with exos. This produced a significant decrease in primordial follicles and a significant increase in late antral follicles compared to age-matched controls, which showed that exosomes induce follicular maturation. Knowing this, another batch of 10-month-old mice were injected with UC-MSCexos or PBS. The mice were mated with adult male mice. The mean number of pups per female was 2.5-fold higher and the mean number of females that did not deliver was 2-fold lower in the exo group compared to the control. Here, we can see evidence of the qualitative effect of exo administration; the exo group had 2 times fewer abnormal spindles, a significant decrease. The exo group oocytes also had significantly lower ROS levels and significantly higher mitochondrial activity. When looking together at the PI3K/MTOR pathway activation and the above ovarian improvements, UC-MSC exosomes appear promising in ovarian rejuvenation for developing fertility treatments in peri/post-menopausal women.

#### 4.2.3. Human Placenta-Derived Mesenchymal Stem Cells (hPD-MSC)

A third place where mesenchymal stem cells may be harvested from is the placenta. One recent study looked at the effect of injecting a solution containing mesenchymal stem cells derived from the placenta (hPD-MSC) into rats of advanced age, 52–54 weeks old [63]. The same research group had also looked at hPD-MSC as a treatment for retinal pigment epithelial cells damaged by hydrogen peroxide-induced oxidative stress and found that it had anti-apoptotic properties and decreased mitochondrial ROS levels [74]. Given that its mechanism of action is oxidative stress, it may have potential for the treatment of ovarian oxidative stress.

The rats were either injected once or three times daily, with age-matched controls. Single-injection therapy provided a significant increase in secondary, preantral and antral follicles during the first week post-treatment. This can be interpreted as an initial boost in follicular maturation. Triple injection led to a significant increase in primary follicles in weeks 2 and 3 compared to the control. The researchers understood this to be a sign of the potency of the multi-dose regiment in maturing the primordial follicles and hence later creating higher levels of primary follicles. Any significant differences compared to the control lasted less than 3 weeks. However, this may be due to the duration of the injection’s effects. While the researchers were perhaps more encouraged by the results of the triple dosage, it is the single dosage that had a significant increase all the way to the antral follicle stage, a good marker for potentially increasing fertility. AMH levels increased significantly for both injection therapies. The treatment appeared to sustain AMH levels, whereas age-matched control levels declined each week, suggesting improved ovarian function in the hPD-MSC groups. The researchers analysed circulating miRNA levels associated with ovarian aging and found an upregulation in three associated negatively with aging (i.e., they increase with aging: miR-21-5p, miR-132-3p, miR-212-3p) and a downregulation in four associated positively with aging (i.e., they decrease with aging: miR-16-5p, miR-34a-5p, miR-145-5p, miR-191-5p). The study was used primarily to examine if hPD-MSC transplantation during ovarian senescence could ameliorate menopausal symptoms and consequently used menopausal rats. However, looking at the above articles on MSC transplantation, the results look promising enough to encourage research in younger, perimenopausal rats, as the procedure could extend their fertility window.

#### 4.2.4. Menstrual Blood-Derived Mesenchymal Stem Cells

Wang et al. [66] looked at a final set of mesenchymal stem cells, this time derived from menstrual blood (MenSC). Female aged mice were injected with MenSC and examined 8 weeks later. The study group had a significantly lower number of atretic follicles when compared to the control, as well as a significantly higher number of antral follicles. Additionally, the study group had significantly higher levels of mtDNA in the ovaries, suggesting increased mitochondria activity. In IVF, the group also had a significantly higher blastocyst formation rate when compared to the control. These results suggest that the treatment may aid in natural fertility preservation by lowering the rate of atresia and restoring mitochondrial function. The treatment appears to be useful as a complementary treatment during IVF too, as it could boost success rate.

#### 4.2.5. Human Endothelial Progenitor Cells

Human endothelial progenitor cells (hEPC) are a heterogenous cell group that have proven difficult to define over the past decade. The current consensus is that they can differentiate into endothelial cells and are capable of forming blood vessels [75,76]. A study looked at the effect [64] of IV administration on mouse ovarian aging. Four- and six-month-old mice were injected with a dose of 5 × 10^4^ hEPC cells twice. Compared to their age-matched controls, both groups showed a decrease in pro-inflammatory cytokines Ifn-γ and Il-1β. The 6-month-old group had a significant decrease for Tnfα as well. The relative expression of the anti-apoptotic gene Bcl2 was significantly increased in both groups. However, the number of ovulated oocytes did not show any significant difference between the study and control groups. Given the positive change in the markers looked at above, this indicates a primarily qualitative improvement in the ovaries, the researchers noted. However, to better establish that there is no quantitative improvement, it would have been beneficial for the study to have looked at the ovarian follicular levels and litter sizes. Nonetheless, the study establishes a qualitative improvement through IV administration of hEPC delaying ovarian qualitative decline.

#### 4.2.6. Adipose-Derived Stem Cell Conditioned Medium (ASC-CM)

Adipose-derived mesenchymal stem cells (ASC) are stem cells derived from adipose tissue [77]. They can induce angiogenesis, inhibit skin ageing and have antioxidant effects on surrounding tissues [78]. A conditioned medium (CM) derived from ASCs has been shown to contain the antioxidant factors released from ASCs and as such, its administration may have healing properties. Using the medium has advantages such as easier and faster production and transportation, and it does not have the risk of the immune system rejecting it [79]. One study looked at its effects on aging mice [65] (Figure 4). The ASCs, which were donated from a 39-year-old healthy woman, were cultured for 72 h in two different media. The final conditioned medium would be collected during administration. Mice were either 4 months or 6 months old, with each age group being split into a control and study group. The study groups either received an injection three times at eight-day intervals (8D-3T) or six times at four-day intervals (4D-6T). Then, 1–2 mice per group were mated with a male post-experiment. The 6-month-old 4D-6T mice had significantly higher numbers of implanted foetuses compared to their age-matched controls. The 4-month-old mice did not show any difference. This suggests that the effects are more noticeable with increased age, when oocyte deterioration is higher. Furthermore, the 4-month-old pregnant mice (both groups) had a significant decrease in ovarian caspase-3, suggesting lower apoptosis. Similarly, both interval groups of 4-month-old mice had a significant increase in Sod2 and Gpx1, two antioxidant-related genes. Catalase, another antioxidant-related gene, increased significantly only in the 4D-6T group. With regard to the 6-month-old group, the ovaries had a significant increase for Bcl2 and Gpx1 in both interval groups, with Bcl2 being an anti-apoptosis-related gene. Like the 4-month-old group, only the 4D-6T group had a significant increase in catalase. From these results, it is evident that ASC-CM administration has antioxidant effects on the ovaries. Stem cell conditioned media could be an interesting alternative to stem cell transplants in increasing fertility as they would be cheaper, more readily available and safer for the patient.

#### 4.2.7. Autologous Mitochondrial Transfer

Autologous mitochondrial transfer (ADSC) involves taking mitochondria from one source and injecting them somewhere else where they are needed, all in the same patient. In one study [80], adipose stem cell-derived mitochondria were extracted from subcutaneous abdominal fat and then injected into collected oocytes. These oocytes would then be used for intracytoplasmic sperm injection, a type of assistive reproductive treatment. The study found that oocytes with mitochondrial microinjections had a significantly higher rate of embryogenesis compared to the control. Furthermore, once the embryos were transplanted back into the mice, the mitochondrial group had a higher number of total pups born compared to the control, although there was no significant difference. Therefore, ADSC mitochondrial transfer may act as a fertility treatment for aging patients with a lower risk profile compared to established methods.

#### 4.2.8. Mitochondrial Transfer from Endothelial Mesenchymal Stem Cells

Zhang et al. [67] examined the effect of mitochondria transfer to aged mice from endothelial mesenchymal stem cells. In their experiment, 10-month-old mice oocytes were collected and injected with mitochondria in vitro (Figure 5). Besides significantly increasing the membrane potential of the oocytes, the mitochondrial injection also significantly lowered their rates of aneuploidy and significantly increased the percentage of normal spindles when compared to age-matched controls. In IVF that was performed afterwards, embryos subjected to the mitochondria had significantly higher embryo formation compared to embryos that were untreated. Once transplanted into mice, the mitochondria-treated embryos had significantly higher birth rates per embryo injected. The results suggest that the mitochondrial transfer helped in the upkeep of cellular integrity by maintaining the high ATP demand required in oocytes. The procedure may have potential as either a preventative or treatment option for maintaining/restoring fertility, though its methodology is quite invasive.

### 4.3. Caloric Restriction and Caloric Restriction Mimetics

Three studies were identified in caloric restriction and its mimetics for delaying ovarian aging (Table 3).

#### 4.3.1. Caloric Restriction Diets

Caloric restriction is under study for its potential beneficial actions on the body when it comes to aging, such as reduced DNA damage and oxidative stress [84]. It may provide benefits in terms of inhibiting ovarian aging as well.

Luo et al. looked at the effect of just CR on rats [81]. The rats were placed under either 25 or 45% caloric restriction compared to a control group that was fed ad libitum. There was also a group fed a high-fat diet. Compared to the control, both CR groups had a significantly higher percentage of primordial follicles (from total) and a significantly lower percentage of atretic follicles. The number of primordial follicles was, in fact, 2-fold higher compared to the control for both groups. The CR diet clearly prevented the maturation of primordial follicles without increasing their rate of atresia either, therefore preserving the ovarian reserve. Curiously, only 25% CR had a significant increase in SIRT1 and SIRT6. As we shall see in the studies below, there appears to be a certain range in which CR inhibits reproductive aging. Higher values may cause malnutrition and limit any acquired benefits. The study does show a link between CR and a higher follicular reserve [81]. It would have been interesting if the study had mated the mice after the caloric restriction period to observe whether the mice would suffer any secondary reproductive damage; the caloric restriction could, for example, have caused damage outside of the follicular reserve that would render the mice functionally infertile. However, in a similar experiment described below, Isola et al. [82] showed that it is unlikely, given that mice did not have trouble birthing post-caloric restriction.

Isola et al. [82] examined the effect of a low level of caloric restriction. Mice were placed under either 10% or 30% CR. The advantage of this study is the 6-month study period, longer than the other two studies mentioned. Interestingly, whilst the primordial follicles were significantly higher for the CR10 group compared to the control, the CR30 group had a non-significant increase. This is not in line with what we would expect from the studies mentioned above. The researchers noted that this may be related to the methodology of using follicular density rather than total follicular quantity measurements. The CR30 group did have a significantly higher pregnancy rate compared to the control, which shows that most likely, the primordial density does not reflect the higher primordial count compared to the control. There was no significant change for the CR10 group compared to the control. It also shows that CR does not cause secondary reproductive damage that would reduce pregnancy. Overall, CR did seem to prolong ovarian longevity, though it was unclear to what extent CR10 was beneficial.

To conclude, there does not seem to be a linear correlation between CR and fertility preservation. Instead, there seems to be an optimal CR percentage range that enables the maximum fertility preservation without triggering malnutrition on one end of the scale, and without causing insufficient CR to trigger the protective mechanisms on the other end. Diet may also play an important role.

#### 4.3.2. CR and Rapamycin

Garcia et al. [83] looked at the effect of CR and rapamycin usage in mice. The mice were split randomly into three groups: a rapamycin group, a 30% CR group and a control group. The control and rapamycin group received food ad libitum. The CR group was only provided with 70% of the caloric value of the food that they would consume ad libitum; hence, they were given a 30% caloric restriction. The rapamycin group received 4 mg/kg every other day during the whole experiment. As expected, the bodyweight and the amount of visceral fat were significantly lower by the end for the CR group compared to the control. Rapamycin is a CR mimetic [85,86] that inhibits the mTOR signalling pathway and has been found to increase mouse lifespan, though the exact mechanism remains unclear [87]. It did not create any significant difference in the study groups compared to the control. Both study groups had a significantly higher number of primordial and tertiary follicles compared to control and both had a significantly lower number of primary follicles. Neither the number of transition follicles nor the number of total follicles showed any difference compared to the control in either group. It would have been interesting to see the number of atretic follicles in this study, but this was not measured. The only significant difference in gene expression was for Foxo3a, both groups having relative levels significantly higher than the control. This was expected as Foxo3a is a sign of increased ovarian reserve and decreased primordial follicle activation [88,89]. Therefore, due to the similarity in results between the two research groups, rapamycin has the potential to inhibit ovarian aging by inducing an ovarian state like that found in CR. Rapamycin inhibits primordial follicle activation and may be an easy intervention to prolong female fertility.

### 4.4. Other Treatments

Finally, five studies were found on other treatments not fitting in the above categories (Table 4).

#### 4.4.1. DHEA

Dehydroepiandrosterone, or DHEA, is a steroid hormone that has become popular as a supplement for its claimed effects of anti-ageing and anti-inflammatory properties and increasing overall wellbeing, among other claims [95]. Given that it is the precursor to sex hormones, its age-related decline has been hypothesised to influence fertility. One study looked at DHEA supplementation in mice near menopause [90]. The study indeed found that the group of mice supplemented with DHEA had significantly higher primordial, primary and secondary follicle counts compared to the control. In a similar fashion, atretic follicle count and DNA-damaged follicle count were both significantly lower in the study DHEA group compared to the control. The study stated that DHEA “partly alleviated the age-related decline in cohesin levels”. Indeed, the quantities of all cohesin subunits tested (REC8, SMC1β, SMC3) were higher in the DHEA group compared to the control, showing that degradation occurred at a slower rate during the timespan of the experiment in the study group compared to the control group. However, the study did not discuss how the partial decrease was not statistically significant. Nonetheless, the results for follicle counts suggest that DHEA promotes folliculogenesis and inhibits both follicular atresia and DNA damage.

#### 4.4.2. Hyperbaric Oxygen Treatment

Hyperbaric oxygen treatment (HBOT) involves the administration of enriched oxygen for a certain time period. Ma et al. [91] studied its effects on the ovaries of aging mice. After 10 consecutive days with 90 min of HBOT administration in aged female mice, the HBOT group of mice had significantly increased AMH concentrations compared to their age-matched controls. After the mice were sacrificed, it was revealed that the number of atretic follicles was also significantly lower. Oocytes were extracted for IVF, and the HBOT group had significantly more oocytes extracted with significantly higher blastocyst formation as well once the samples were fertilised. Therefore, this treatment may have benefits as a minimally invasive adjuvant to IVF. It would have been interesting to observe the effects of HBOT on natural fertilisation, as it may even have utility outside IVF for promoting fertility.

#### 4.4.3. Macrophage-Derived Extracellular Vesicles

Macrophages play an important role in ovarian protection, folliculogenesis and tissue repair. Xiao et al. [92] looked at the effect of the injection of vesicles from M1- and M2-type macrophages. The group suggested that vesicles derived from such macrophages may have beneficious effects, as they found them to contain microRNAs (miR-107 from M1-derived macrophages and miR-99a-5p from M2-derived macrophages). The injection of M2-type vesicles into mice over a period of 4 weeks led to a significantly higher count of primordial follicles when compared to the age-matched control group. Interestingly, the percentage of growing follicles in the M2 group was significantly lower than in the control group. The M1 group showed opposite effects to M2; it had a significantly lower number of primordial follicles compared to the control while having a significantly higher number of growing follicles. This suggests that M1 works by inhibiting folliculogenesis while M2 promotes it. Indeed, in a Western blot analysis, p-AKT (downstream of PI3K pathway) and p-RPS6 (downstream of mTOR pathway) had significantly higher relative intensity in the M1 group compared to the control, while they were significantly less intense in the M2 group compared to the control. mTOR, as we have seen, promotes folliculogenesis when activated and lessens it when inhibited. In IVF, the M2 group had a significantly higher number of retrieved oocytes, while the M1 group did not display any significant difference. The results suggest that the macrophage-derived extracellular vesicles have potential as a treatment for either promoting or limiting folliculogenesis. Therefore, they may find usage in both ovarian aging prevention (M2 usage) and treatment during IVF (M1 usage) according to the type of macrophage used.

#### 4.4.4. Cell-Free Fat Extract

Cell-free fat extract (CEFFE) is a novel experimental treatment method that is currently being researched primarily in China for applications such as osteoarthritis attenuation [96], tissue regeneration [97] and osteoclastogenic modulation [98]. It is rich in growth factors and anti-inflammatory properties, both of which have had positive effects in other studies mentioned in this review. Liu et al. [93] examined its effects on aged mice. CEFFE was extracted from adipose tissues. 10-month-old female mice were injected with 200 μL CEFFE at a protein concentration of 3 μg/mL every other day for two weeks. After the treatment, some were made to superovulate and mated with young male mice. The mice were then either sacrificed in order to examine the ovaries or allowed to give birth. Two months after the injections finished, CEFFE-injected mice, when compared to their age-matched controls, had significantly higher AMH values as well as a significantly higher number of primary, secondary and total follicles. In the sacrificed mice, a significantly higher blastocyst formation rate percentage was noted, while the non-sacrificed mice had a significantly larger litter number, with around twice as many pups per birth. It would have been interesting to examine the levels of anti-inflammatory markers in the bloodstream. The researchers hypothesised that some of the anti-aging effects were derived from DNA structural restoration. Indeed, γH2AX relative staining levels (a biomarker for double-strand breaks [99]) in the ovaries were significantly lower in the study group than in the age-matched controls. Overall, these results suggest that CEFFE was able to restore ovarian function in aged mice and could be useful in preventing ovarian aging.

#### 4.4.5. TrkB Agonists

The brain-derived neutropic factor (BDNF) has endocrine properties in the ovaries and plays an important role in their development. Its receptor is TrkB, which may be targeted with antibodies. Through BDNF gene knockout mice, it has been demonstrated that it plays an important role in the maintenance and maturation of ovarian cells, while it was originally assumed that their role was primarily played in neuronal cells [100]. TrkB activation triggers the PI3K/AKT/mTOR pathway, the Ras/MAPK pathway and the PLCγ1 pathway, all playing important roles in ovarian function [101]. BDNF serum levels decrease with age [102], so targeting the TrkB receptor in aged ovaries may rejuvenate their function. Qin et al. [94] looked at the TrkB antibody Ab4B19 in aged mice. The ovaries of aged mice treated with the TrkB agonist had significantly higher primordial, preantral and antral follicle counts compared to their age-matched controls. At the same time, the study group also had significantly lower atretic follicle counts. The mice that were allowed to mate also produced significantly more ovulated oocytes per mouse and had a significantly larger mean litter size compared to the control. The results suggest that the agonist both enhances fertility and preserves the follicles. The researchers also tested a group of mice treated with cyclophosphamide to induce ovarian damage and found that the application of Ab4B19 repaired and restored the quality of the oocytes. The above results, in combination with its low adverse effect profile in mice, make it a candidate for a potential future pharmaceutical to prevent ovarian aging, as well as to treat infertility.

## 5. Discussion

As the proportion of women opting to have fewer children and at a later age is increasing, the problem of ovarian aging is becoming urgent. There are, however, factors that can be addressed at the molecular level to extend the ovarian reproductive timespan. These factors should be addressed by examining the mechanisms leading to ovarian aging and how they may be mitigated by looking at the latest data on preventative treatments.

Not all ovarian aging factors are created equal; indeed, some can be modified, while others are too intrinsic for us to currently be able to address them. The latter include genetic factors that determine the exact rate of menopause and, hence, the genetic maximum age of menopause for an individual. If we were to think of the age of menopause as being determined from a sum of all ovarian aging occurring through different mechanisms, inherited genetic factors would account for roughly half of the total [103]. Even though there is little to be done to address them, some potential may be found in using AMH and other fertility biomarkers to predict the age of menopause, though its usage on an intraindividual basis is still debated [104].

It is still unclear whether telomere length, which decreases with each cellular division, can be effectively addressed with telomerase activators, and as such, they are not being administered medically. Shorter telomere length is associated with aging, and given that a shorter length may, at some point, lead to atresia, as discussed above, telomerase activators could be capable of prolonging the female fertility window. There have been some studies looking at telomerase activators and longevity [105]. However, their effect on ovarian fertility has not been well explored, which would be an interesting point of study. DNA mutation, such as on BRCA1, is another type of factor that we do not currently possess the means of countering. Knowing if someone is a BRCA1/2 carrier can, however, be useful for exploring ART options, which would aid in fertility.

There are, however, other factors that can be prevented to a certain extent, namely oxidative stress, cohesin deterioration, DNA damage and rate of atresia. Of the four, oxidative stress the easiest to address in theory, by taking up more antioxidants, which would act as ROS scavengers, mitigating the latter’s deleterious effects. The best source of antioxidants seems to be a healthy and balanced diet [106]. As oxidative stress’ downward effects influence all other mechanisms, any reduction in its levels would also inhibit aging through all other mechanisms. The alleviation of preventable factors may prolong the ovarian aging window by a few years, which would have a large impact on the fertility rates at a societal level.

Medical treatments as a prevention tool for ovarian aging appear to have three common methods used to prolong ovarian aging. The first is inhibiting the rate of follicular activation and thus keeping a larger number of primordial follicles over time, the second is inhibiting the rate of follicular atresia and the third is preventing structural damage of the ovarian cells, such as DNA or cohesin deterioration. The treatments are either prescription pharmaceuticals or more invasive treatments such as stem cell injections. Hyperbaric oxygen treatments may be their own third category, as they are minimally invasive.

Stem cells are an exciting new field in ovarian aging that could find many uses. Currently, their primary drawback is that most types being examined (hAMSC, UC-MSC, hEPC, ASC-CM, MenSC) require injections. It would be interesting to explore the duration of their ovarian rejuvenating effects. Perhaps some have lasting effects that would make them viable for usage as a preventative of ovarian aging. Two initially promising stem cell treatments that we examined are ASC-CM and UC-MSCexos. Both work by using stem cells indirectly, which comes with various advantages such as cost savings, a better safety profile and fewer ethical considerations. The problem with UC-MSCexos is that its primary mechanism of action seems to be through the activation of the mTOR/PI3k pathway. As this triggers folliculogenesis, it is unclear if it could be used as a prevention technique. In a similar manner, ASC-CM also appears to have a stronger potential as a treatment, given its effects on birth rates. However, ASC-CM seems to derive its benefits primarily from its antioxidative actions and could, therefore, have applications as an ovarian aging preventative. Further research into the duration of its rejuvenating effects is necessary. We looked at four direct stem cell treatments: hAMSC, hPD-MSC, MenSC and hEPC. The first two significantly increased the ovarian levels of the fertility biomarker AMH. Between these two, hAMSC gave stronger results in terms of preventing follicular atresia. The third treatment, MenSC, appeared to be targeted to restoring mitochondrial function. In the future, it may find usage in combination with other stem cells treatments as an ovarian rejuvenation therapy during or prior to IVF. The final treatment, hEPC, was one of the few anti-inflammatory treatments. It did not affect the number of ovulated oocytes, so its effects appear to be mainly qualitative. All four of the above required injections and, as a result, would not be ideal in a long-term, preventative scenario.

The mitochondrial transfer treatment was another type of novel treatment that may have future applications. In the studies of Wang et al. [66] and Zhang et al. [67], the treatments required a direct injection of mitochondria into oocytes. As a result, it is not currently feasible as a preventative of ovarian aging on a wide scale. In the studies, mitochondrial injections were found to have a topical effect in the injected cells, and they could find usage in IVF treatments.

The TrkB agonist treatment [94] with the antibody Ab4B19 presented interesting results as it displayed both lowered atresia rates and higher primordial, preantral and antral follicle counts. However, any antibody treatment does have the potential to attack other, currently unknown, sites. Therefore, further animal research, specifically on its safety profile, will be necessary before it can be considered for human trials.

Hyperbaric oxygen treatment has been shown to have effects such as upregulating antioxidant gene expression in endothelial cells [107] promoting angiogenesis in skin cells [108] and increasing telomere length in blood cells [109]. It is unclear how HBOT exerts its synergistic effects; the higher oxygenation of tissues must have downstream effects on some beneficial pathways. The results from Ma et al. [91] suggest that HBOT may preserve the quality and quantity of oocytes. More research is necessary to explore the duration of positive effects of treatment, the mechanism of action and how it may be implemented in ovarian aging prevention or IVF treatments. This type of treatment is of particular interest as it is the least invasive medical treatment currently being studied.

Cell-free fat extract is another novel treatment with a lot of recent research on its benefits in terms of regenerating tissues and attenuating chronic diseases [96,97,98]. Its positive results in the study by Liu et al. [93] combined with its low risk for adverse effects make it an interesting candidate for further study. It appeared to have DNA repair mechanisms and as such, further research should focus on the exact mechanism of repair, which the authors were not able to identify in this study.

We examined four potential pharmaceutical preventative treatments: Visfatin, S1P, DHEA, metformin. Visfatin activates the mTOR/PI3K pathway, which eliminates it as a potential preventative. The S1P study can only serve as an indication of future potential, and it is currently unsuitable for ovarian aging prevention. The study’s largest drawback was that the human samples were from human endometrioma patients. The researchers noted that this would make it unclear to what extent its results can be extrapolated to predict its effects in healthy individuals. The metformin study gave more reliable results as it was carried on for the relatively long period of 6 months, and the treatment was well tolerated and decreased the rate of atresia. Given that its already established as a rather safe medication for diabetes [110], it could have potential as a preventative. However, giving diabetic medication to non-diabetic patients could cause hypoglycaemia. Research is necessary on the systemic effects of metformin injection near the ovaries before it is to be considered. Finally, DHEA and androgens require a fine balance to prevent hyperandrogenism. Chu et al. [90] mentioned that the androgenic effects of DHEA are rather weak compared to testosterone. However, we would still caution against their use in young patients. DHEA is a pre-hormone, presenting less of a risk than outright hormonal treatment, and seems to be relatively safe in short-/medium-term usage [111,112,113]. Nevertheless, there is a lack of studies on its long-term supplementation safety profile. Given the above, we believe that DHEA will only find limited usage as an ovarian aging treatment and should not be considered as a preventative.

A final treatment is a moderate caloric restrictive diet and CR-like phrmaceuticals such as rapamycin. Although obesity on its own affects fertility, the studies analysed show that it also has potential in mice (and subsequently patients) of a healthier weight. Moderate CR should not cause any secondary damage to the reproductive system as the risk of malnutrition is limited. It is an option that could be sustainable on a long-term basis and requires minimal medical assistance, which makes it appealing. In a similar manner, rapamycin would also be an easy method to induce a CR-like state, given that it is better established to be well tolerated for non-diabetic patients.

The treatments examined provide an analysis of various methods for how ovarian aging may be prolonged. The age of menopause, which was thought to be a hard limit on the age of reproduction at around 45 years of age, may be more flexible than initially thought, and may have the prospect of being extended. Our analysis shows that further research is needed if this is to be materialised, focusing primarily on human subjects for the better-tolerated treatments such as supplements and further animal studies for the treatments that are more experimental.

## 6. Conclusions

We suggest such interventions to be targeted to stem cell research, mitochondrial transfer and CR-like medicine, as they are the most likely to provide a breakthrough. First, stem cells are recommended as all the types examined appeared efficacious as an ovarian aging preventative. New sources of stem cells are being discovered, and the potential to source them directly from the patient makes them a low-risk treatment. Second, mitochondrial transfer is promising as it directly targets the problem of mitochondrial aging, which, as we established, is amplified in ovarian cells due to their increased energy demand. If a less invasive procedure is established, with direct in vivo injection, it could prove useful on a large scale. Third, CR-like medicine is recommended as caloric restriction is one of the better-established longevity-inducing mechanisms. Its potential to trigger the physiological CR mechanism without the necessity of a constant CR diet could make it an easy and accessible preventative mechanism of ovarian aging. The treatments above may have synergistic potential if stacked together. However, their individual mechanisms should first be demonstrated in humans before such interactions are examined. It is also important that studies investigate the longer-term effects of these treatments, at the very least with a duration of multiple months. Further research may allow for developing fertility preservation methods that are effective before fertility assistance treatment is necessary.

## Figures and Tables

**Figure 1 ijms-24-09828-f001:**
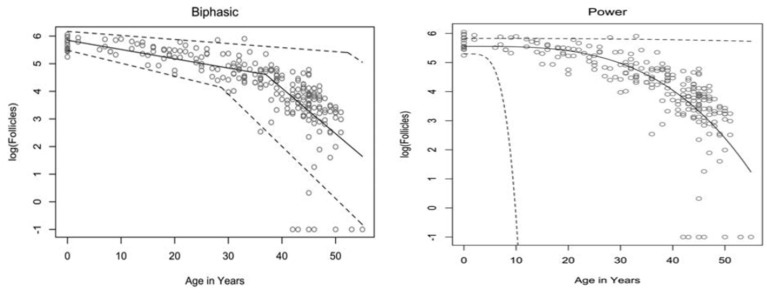
Models of human follicular loss over years, as seen in Coxworth et al. [28], with permission of Oxford University Press. Model represented by solid line, confidence intervals by dashed line.

**Figure 2 ijms-24-09828-f002:**
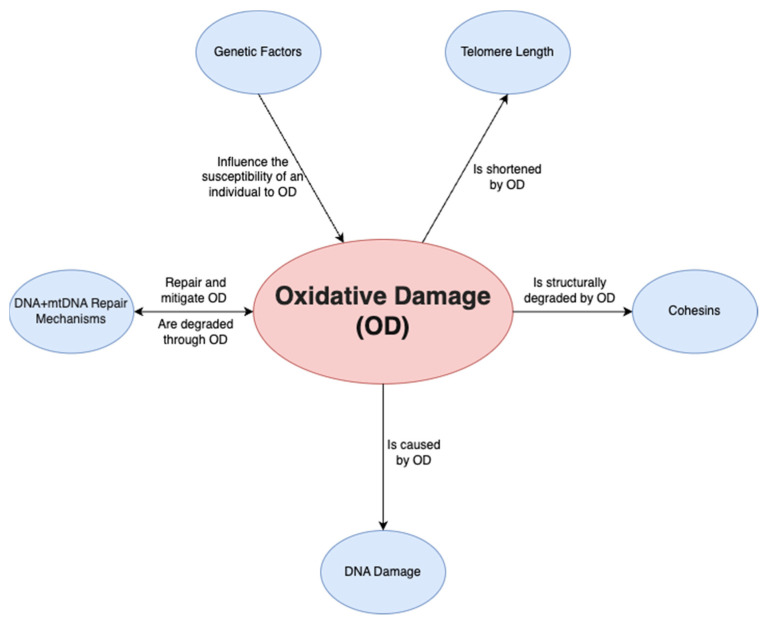
Relationship of oxidative damage to other mechanisms.

**Figure 3 ijms-24-09828-f003:**
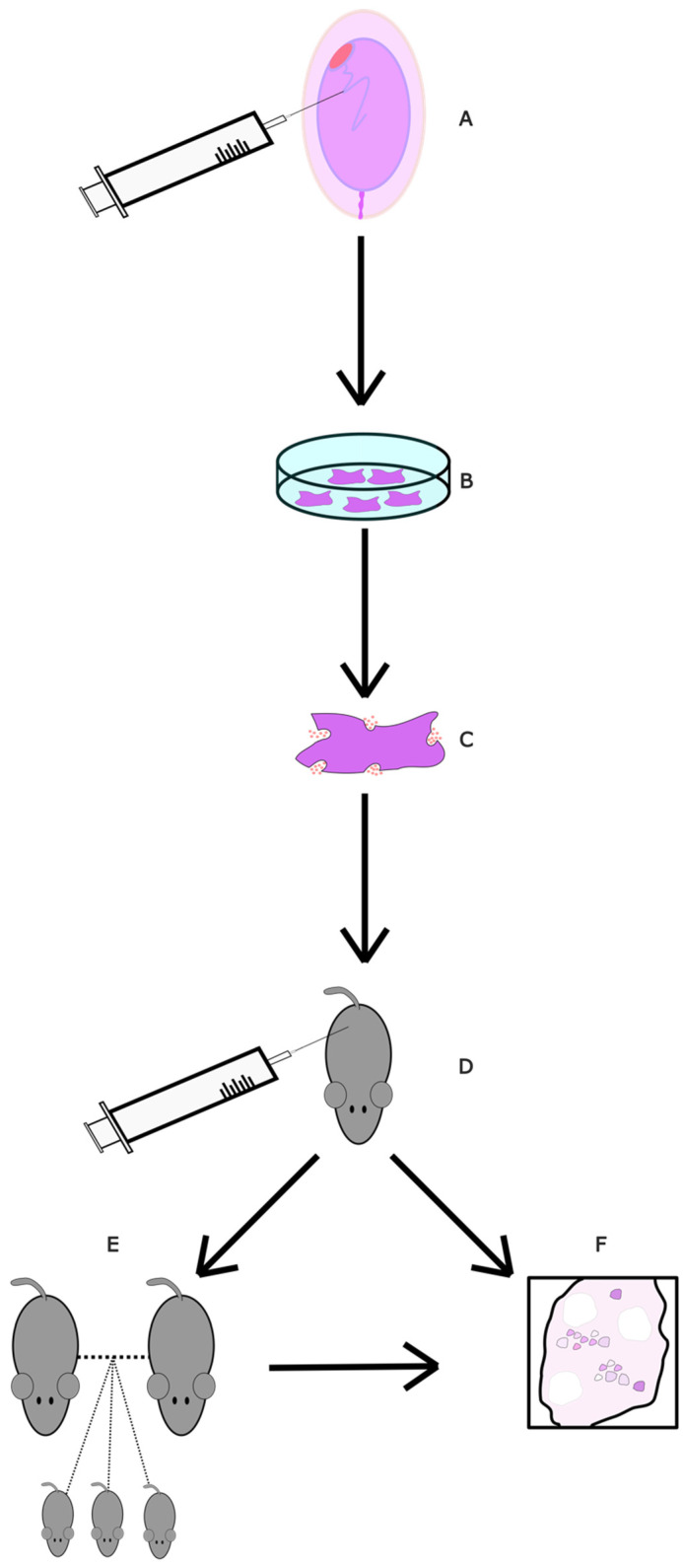
Diagram of UC-MSC exos study. (**A**) Stem cells from umbilical cords of consenting mothers were collected and isolated. (**B**) UC-MSCs were cultured. (**C**) UC-MSCexos were isolated through centrifuging and filtering. They were then diluted in PBS. (**D**) UC-MSCs were injected into the ovarian bursa. As a control, some mice had one bursa injected with just PBS and the other with UC-MSCexo dilution. (**E**) A group of mice was allowed to mate for a period of 4 months, with numbers of offspring recorded. Afterwards, they were euthanised for follicular analysis. (**F**) Finally, 3 weeks post-injection, a group of mice was euthanised and had their ovaries collected for follicular examination.

**Figure 4 ijms-24-09828-f004:**
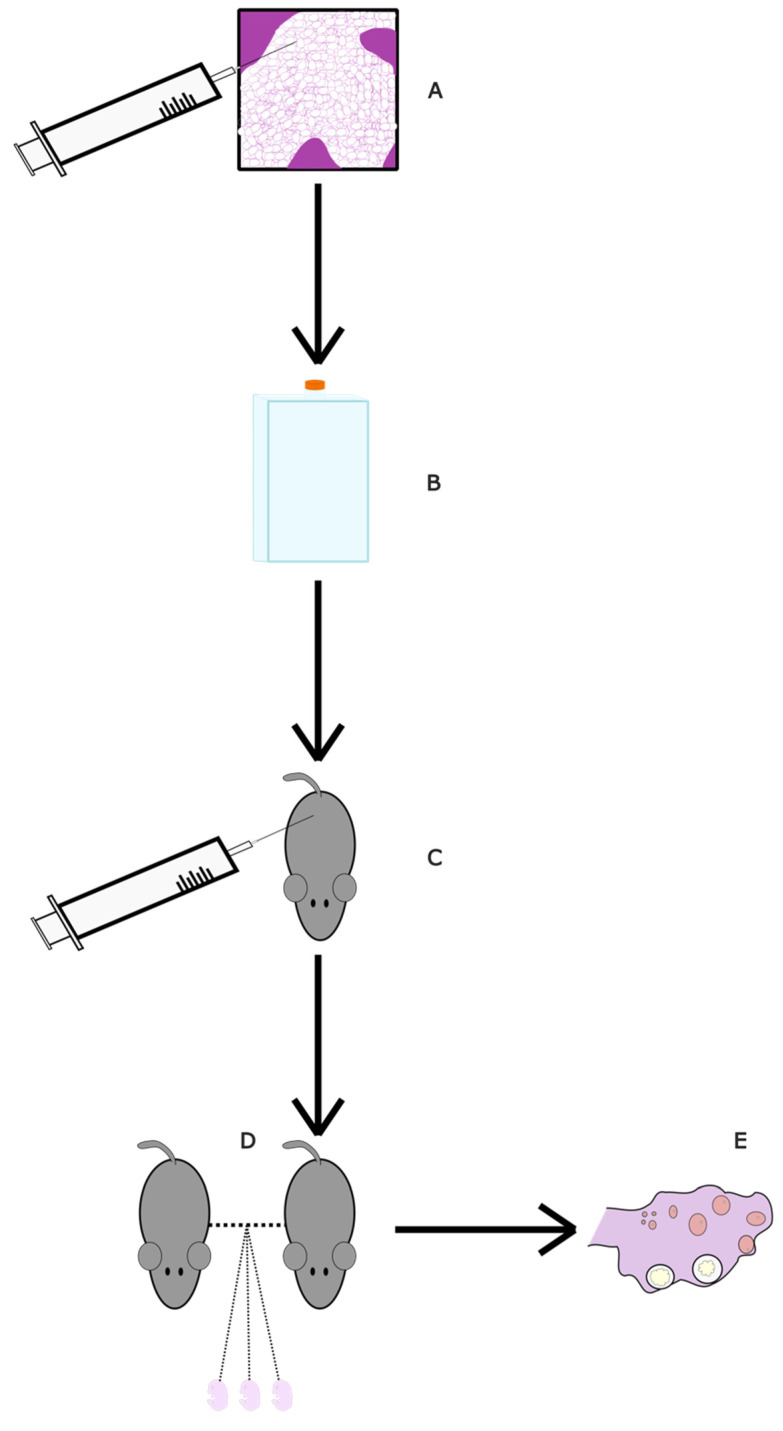
Diagram of ASC-CM study. (**A**) Adipose stem cells were collected from a consenting donor and isolated. (**B**) ASC-CMS were thawed and then sub-cultured in a hyperflask with AMSC medium, which was afterwards replaced with DMEM. The culture medium was collected and replaced with fresh medium five times in order to collect ASC-CM. (**C**) Mice were injected in the tail vein according to their assigned group: control, 8D-3T, 4D-6T (see Table 2). (**D**) The mice were allowed to mate with male mice until pregnant. (**E**) Six days post-injection, the mice were euthanised. The number of implanted foetuses was recorded. Ovaries were analysed for gene expression related to oxidative damage and apoptosis.

**Figure 5 ijms-24-09828-f005:**
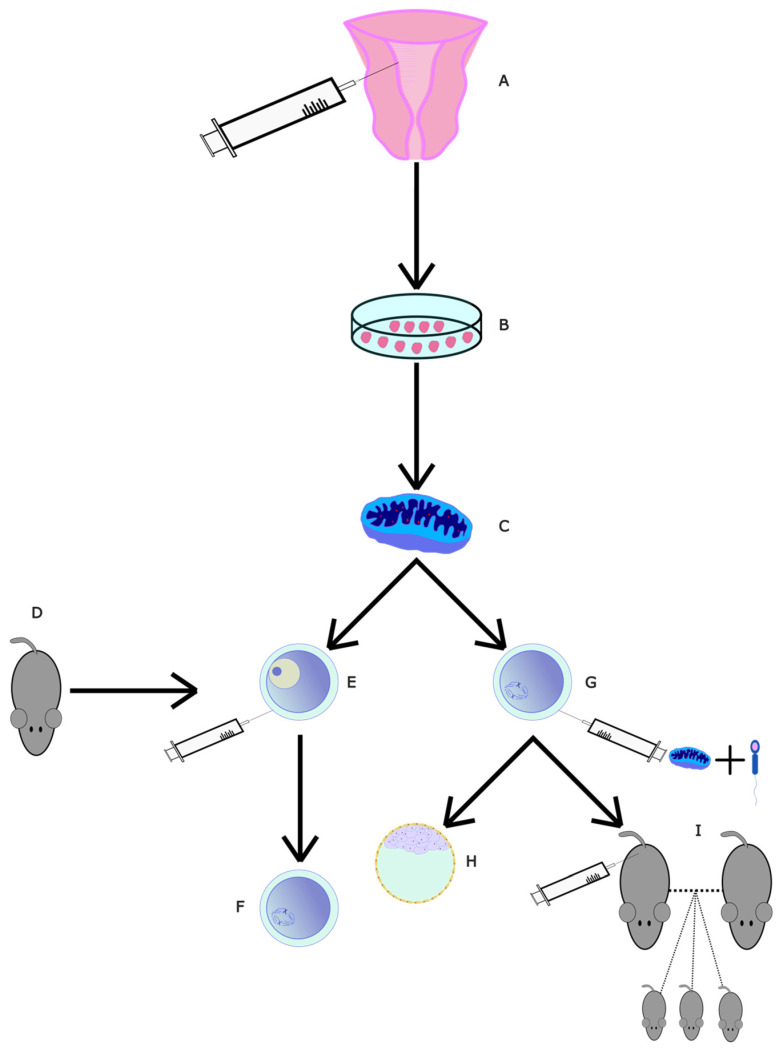
Diagram of mitochondrial transfer from EN-MSCs. (**A**) EN-MSCs were collected and isolated from 10-month-old mice. (**B**) EN-MSCs were cultured. (**C**) Mitochondria were extracted and isolated. (**D**) Then, 10-month-old mice had (**E**) GV-stage oocytes and (**G**) MII-stage oocytes collected. (**E**) GV-stage oocytes were micro-injected with mitochondria and allowed to develop into (**F**) MII-stage oocytes. (**G**) In MII-stage oocytes, intracytoplasmic sperm injection (ICSI) was performed alongside mitochondrial microinjection. One group of these fertilised oocytes was allowed to mature into (**H**) embryos. Another group was (**I**) injected into surrogate mothers to perform IVF.

**Table 1 ijms-24-09828-t001:** Studies on pharmaceuticals.

Author	Pharmaceutical	Design and Method	Ovarian Effect	Conclusions
Qin et al., 2019 [50]	Metformin	Mouse study: Female 28-month-old mice given 100 mg/kg metformin (MET) for 6 months, repeated with a control group	MET had higher primordial follicle count (MET ~10 vs. control ~5, *p* < 0.01), primary follicle count (MET ~60 vs. control ~40, *p* < 0.05) and SIRT-1 levels (MET ~1.6 vs. control 1.0, *p* < 0.05). MET had lower 8-OHdG, (MET ~30 vs. control ~40, *p* < 0.05), 4-HNE (MET ~30 vs. control ~50, *p* < 0.05) and p16 (MET ~30 vs. control ~40)	Metformin delays ovarian aging in mice
Park et al., 2020 [51]	Visfatin	Mouse study: 18-month-old mice were injected with 0.1 mL 500 ng/mL Visfatin or 1000 ng/mL three times at two-day intervals. Experiment 1: 12 mice split into 0 (control), 500 or 1000 ng/mL Visfatin groups. Experiment 2: 12 18-month-old female mice split into two groups: 500 ng/mL Visfatin and 1000 ng/mL. The mice were mated with males	Experiment 1: number of primary, secondary and early antral follicles higher than control in Visfatin groups. For 500 ng/mL: (primary 26.0 ± 1.5 vs. control 14.0 ± 1.5, *p* < 0.05), (secondary 11.0 ± 0.6 vs. control 6.0± 1.0, *p* < 0.05, antral 6.0 ± 0.5 vs. control 1.0 ± 0.6, *p* < 0.01). For 1000 ng/mL: (primary 26.0 ± 2.1 vs. control 14.0 ± 1.5, *p* < 0.05), (secondary 10.0 ± 0.6 vs. control 6.0 ± 1.0, *p* < 0.05, antral 4.0 ± 0.6 vs. control 1.0 ± 0.6, *p* < 0.05). Atretic follicles lower than control for Visfatin groups (500 ng/mL 6.0 vs. 13.0 control, *p* < 0.05, 1000 ng/mL 7.0 vs. 13.0 control, *p* < 0.05). Experiment 2: 500 ng/mL had more pups per female (2.20 ± 1.30 vs. control 0, *p* < 0.05)	In mice, Visfatin may recover ovarian function and fertility for post-menopausal aged mice
Guzel et al., 2018 [52]	gS1P	In vitro study: Ovarian cortical tissue from patients. The samples were incubated with S1P concentrations of either 0 (control), 200 or 400 μM for 4 days	Primordial and secondary follicle counts higher in 400 μM and 200 μM vs. control. For 400 μM: (primordial: 2.3 ± 0, primordial control 0.8 ± 0.2, *p* < 0.01), (secondary: 0.7 ± 0.1, control 0.1 ± 0.05, *p* < 0.05). For 200 μM: (primordial 200 μM 1.7 ± 0.2, control, 0.8 ± 0.2, *p* < 0.05), (secondary 200 μM 0.4 ± 0.1 vs. control 0.1 ± 0.05, *p* < 0.05). Number of follicles with positive cleaved caspase 3 marker lower in 200 μM than control (0.8 ± 0.05 vs. control 1.2 ± 0.1, *p* < 0.05) and in 400 μM as well (0.2 ± 0.0.5 vs. control 1.2 ± 0.1, *p* < 0.01)	Under in vitro conditions, S1P inhibits follicular apoptosis and improves survivability in the human ovary
Mumusoglu et al., 2018 [53]	S1P	Rat study: 30 female 10-month-old rats divided into three groups: (A) 0.1 mg/kg Fingolimod (S1P analogue), (B) 1 mg/kg Fingolimod, (C) control. Duration: 60 days	B had higher AMH values (Group B 5.72 ± 0.61 ng/mL vs. Group C 4.81 ± 0.85 ng/mL, *p* = 0.05). A had a higher non-apoptotic ratio (Group A 67.0% ± 16.4% vs. Group C 29.9% ± 19.5%, *p* < 0.001). B had a higher non-apoptotic follicle ratio (Group B; 51.1% ± 11.5% vs. Group C 29.9% ± 19.5%, *p* = 0.023)	S1P may decrease spontaneous follicular apoptosis given its inhibition of the ceramide-induced death pathway

**Table 2 ijms-24-09828-t002:** Studies on stem cells.

Author	Stem Cell Type	Method	Ovarian Effect	Conclusions
Ding et al., 2018 [17]	hAMSC	Mouse study: female mice 12–14 months of age were split into four groups (n = 15 per group and injected with hepatocyte growth factor (HGF), epidermal growth factor (EGF), both or PBS (control). Duration: 4 weeks	The hAMSC group had higher primordial (88% higher, *p* < 0.001), primary (89% higher, *p* < 0.001), secondary (86% higher, *p* < 0.01) and antral follicle counts (81% higher, *p* < 0.001) and AMH than the control (92% higher, *p* < 0.001)	hAMSCs inhibit ovarian aging through HGF and EGF secretion
Yang et al., 2020 [61]	Human umbilical cord mesenchymal stem cell exosomes (UC-MSCexos)	Mouse study: 10-month-old mice were injected with UC-MSCexos. Three weeks later, the ovaries were collected. Another 10-month-old group was injected with either UC-MSCexos or PBS (control)	Primordial follicle % lower in exo group (exo ~14%, control ~17%, *p* < 0.05), late antral % higher in exo group (exo ~12%, ~9% control, *p* < 0.05). Exo group had 2.5× mean number of pups per mouse vs. control (4.5 vs. 2). Exo group % of aberrant spindles 20%, control ~38%, *p* < 0.01. ROS fluorescence intensity in exo group ~20 vs. control ~25, *p* < 0.05. Mitochondrial fluorescence intensity in exo group ~1.6 vs. control 1.0, *p* < 0.01	UC-MSCexos induce follicular maturation and could aid in restoring fertility
Jiao et al., 2022 [62]	Umbilical cord mesenchymal stem cells (UC-MSC)	Mouse study: 9–10-month-old female mice injected with either a) 5 μL MSC-CM + 0.3 mg/mL HA, b) 5 μL MSC-CM + 0.3 mg/mL HA + 1 μg/mL HGF neutralising antibody, or c) 800 ng/mL HGF + 0.3 mg/mL HA	Primary (*p* = 0.046), secondary (*p* = 0.003), antral (*p* = 0.032) and total follicle counts higher in MSC/HA group than control (*p* < 0.05). Higher pups/litter than control (3.33 ± 2.02 vs. 1.6 ± 1.41, *p* = 0.042)	UC-MSC have may prolong natural aging in mice by protecting the follicular reserve
Kim K. et al., 2022 [63]	Human placenta-derived mesenchymal stem cells (hPD-MSC)	Rat study: Rats split into four groups. 1: injected with 0.2 mL PUBS containing 5 × 10^5^ hPD-MSCs. 2: injected with the same dosage three times, at 10-day intervals. 3: A group injected with only PBS (positive control). 4: A group not injected (negative control) Each group n = 24	For single-injection therapy, AMH levels had an increase in week 5 (hPD ~1.5, control 0.75, *p* < 0.05). In week 5, miR-16-5p, miR-34a-5p and miR-191-5p significantly decreased (*p* < 0.05) in hPD-MSC group compared to control (miR-16-5p: control ~1 vs. hPD-MSC ~0.5), (miR-34a-5p: control ~1 vs. hPD-MSC ~0.8), (miR-191-5p: control ~1.1 vs. hPD-MSC ~0.8)	hPD-MSC transplantation improves ovarian function in menopausal rats and alleviates symptoms of ovarian aging
Kim G. et al., 2018 [64]	Human endothelial progenitor cells (hEPC)	Mouse study: 4- and 6-month-old female mice injected twice with 5 × 10^4^ cells at a 4-day interval. hEPCs from four healthy human donors. After being cultured, the hEPCs were injected in a suspension of PBS into each mouse. PBS-only served as control	Ifn-γ and Il-1β both lower in 4- and 6-month-old mice vs. controls. Ifn-γ: (4-month hEPC ~0.25 vs. control 1.5, *p* < 0.01), (6-month ~0.5 vs. control ~1.5, *p* < 0.001). Il-1β: (4-month hEPC ~0.1 vs. control ~1.0, *p* < 0.05), (6 month ~0.25 vs. control ~1.25, *p* < 0.05). Tnfα lower in 6-month-old mice (hEPC ~1.5 vs. control ~2.5, *p* < 0.01). Bcl2 higher in both (4-month ~3 vs. control ~1, *p* < 0.05) (6-month ~1.5 vs. control ~1, *p* < 0.01)	hEPCs may improve ovarian quality with aging
Ra et al., 2020 [65]	Human adipose stem cell conditioned medium (ASC-CM)	Mouse study: ASCs were isolated from a 39-year-old healthy woman. Then, 4- and 6-month-old female mice were randomly divided into control (PBS) and treatment groups. Treatment group given a medium of ASC, either three times in 8-day intervals (8D-3T) or six times in 4-day intervals (4D-6T)	The 6-month-old mice of the 4D-6T group had a higher number of foetuses than the control (4D-6T ~15 vs. control ~7, *p* < 0.05). Ovaries of pregnant mice had lower caspase 3 mRNA levels for both interval groups vs. control (3T-8D ~0.3, 6T-4F ~0.5, control 1, *p* < 0.01 for both)	ASC-CM has antioxidant effects that improve oocyte quality in aging mice
Wang et al., 2022 [66]	Menstrual blood-derived mesenchymal stem cells (MenSC)	Mouse study: 8-month-old C57BL/6J mice were divided into three groups: MenSC injected intraperitoneally, in the ovaries and through tail intravenous injection. MenSC concentration: 6 × 10^5^ MSCs in 50 μL normal saline	At 8 weeks post-transplantation, the MenSC group injected directly in the ovaries had significantly lower levels of atresia compared to the control (~28 vs. 45, *p* < 0.01). The relative mtDNA amount in the ovaries was significantly higher in this group at 40 weeks post-treatment (~2 vs. ~0.8, *p* < 0.05) compared to the control. The MenSC group also had a significantly higher number of antral follicles (~7 vs. ~4, *p* < 0.01). In IVF, the MSC group had a significantly higher blastocyst formation rate (~55% vs. ~35%, *p* < 0.01)	Blood-derived MenSCs may aid in fertility preservation and also assist in IVF as a treatment
Jiao et al., 2022 [62]	UC-MSCs and autocrosslinked hyaluronic acid (HA)	Mouse study: 9/10-month-old C57BL/6J female mice were transplanted with 5 μL UC-MSC + 0.3 mg/mL HA + 1 μg/mL HGF. After 14 days observation, mice were either sacrificed for analysis or allowed to mate	The MSC/HA group had a significantly higher number of total follicles compared to age-matched controls (~500 vs. ~300, *p* < 0.05) and a higher number of primary, secondary and antral follicles (~200 vs. ~100, *p* < 0.05 for all three). Significantly higher number of pups/litter (~4 vs. ~2, *p* < 0.05)	Autocrosslinked HA boosts the effect of UC-MSCs in the ovaries. Together they restore ovarian function in naturally aged mice
Zhang et al., 2023 [67]	Mitochondria supplementation from endometrial mesenchymal stem cells (EN-MSC)	In vitro and mouse study: 10-month-old mice. Mitochondria from EN-MSCs were injected into germinal vesicle-stage oocytes of aged mice	The % of normal spindles in MII oocytes of aged mice was significantly higher in the study group than control (~60% vs. ~40%, *p* < 0.05). The rate of aneuploidy was significantly lower (~40% vs. ~60%, *p* < 0.05). The membrane potential was significantly higher (ΔΨm ~1.25 vs. ~0.4, *p* < 0.0001). When transplanted into mice via IVF, fertilised oocytes also had significantly higher live birth rates (~25% vs. ~12%, *p* < 0.05) and significantly higher blastocyst formation rates as well (~70% vs. ~50%, *p* < 0.05), significantly higher % live birth rates (~25% vs. ~12%, *p* < 0.05)	Mitochondrial supplementation from sources such as EN-MSCs has the potential to restore ovarian function in aged mice and also improve IVF outcomes

**Table 3 ijms-24-09828-t003:** Studies on caloric restriction and caloric restriction mimetics.

Author	Pharmaceutical	Design and Method	Ovarian Effect	Conclusions
Luo et al., 2012 [81]	CR	Rat study: 48 female 2-month-old Sprague Dawley rats were randomly divided into four groups: control, 25% CR group, 45% CR group, high-fat diet group (HF)	The 45CR group had higher levels of primary follicles vs. control (control 19.40 ± 3.99, 45CR 56.71 ± 5.60, *p* < 0.01). The % of primordial follicles was higher for 25CR and 45CR than the control (control 20.5 ± 3.5%, 25CR 41.9 ± 2.9% (*p* < 0.01), 45CR 44.4 ± 2.2% (*p* < 0.001)). The % of antral follicles was lower in the 25CR and 45CR groups than the control (control ~30%, 25 CR ~18% (*p* < 0.05), 45CR ~17% (*p* < 0.05). SIRT1 was higher in the 25CR group than control (*p* < 0.001)	Caloric restriction, especially moderate, maintains the follicular reserve and inhibits ovarian aging
Isola et al., 2022 [82]	CR	Mouse study: 72 female mice were divided into four groups: control, 17α-E2, 10% caloric restriction (CR10) and 30% caloric restriction (CR30). Duration: 24 weeks	Primordial follicle count was higher in CR10 (follicles/mm^2^ CR10 ~3.5 vs. control ~2.8, *p* < 0.05). Transitional follicle count was higher in both CR groups than the control (control ~1.8 vs. CR10 ~2 vs. CR30 ~30). Primary follicle count was higher in CR30 (CR30 ~2.5 vs. control ~1, *p* < 0.05). CR30 had higher pregnancy rate vs. control (100% vs. 44.44%, *p* < 0.05)	CR may prolong ovarian reserve and, thus, increase fertility lifespan in mice
Garcia et al., 2019 [83]	CR and rapamycin	Mouse study: 36 1-month-old C57BL/6 mice were split into three groups: control, rapamycin (rapa), 30% CR. Mice in the rapamycin group received 4 mg/kg rapamycin every other day for 93 days. Afterwards, mice were euthanised and their ovaries were examined	Number of primordial follicles was higher in rapa (*p* = 0.04) and CR (*p* = 0.02) vs. control (rapa: ~12,000 CR: ~13,000, control: ~6000). Number of primary and secondary follicles was lower in both groups vs. control (control ~4000, rapa ~1800 (*p* < 0.05), CR ~1800 (*p* < 0.05) and control ~650, rapa ~300 (*p* < 0.5), CR ~300 (*p* < 0.05)). Number of tertiary follicles was lower in CR (control ~220, CR ~110, *p* = 0.005)	Rapamycin and CR both show signs that they might prolong reproductive lifespan by inhibiting primordial follicle activation

**Table 4 ijms-24-09828-t004:** Studies on other treatments.

Author	Pharmaceutical	Design and Method	Ovarian Effect	Conclusions
Chu et al., 2017 [90]	DHEA	Mouse study: 15 mice were divided into three groups: E2 group, receiving 100 μg/day β-oestradiol. DHEA group receiving 5 mg/day DHEA. Control group receiving saline. Experiment was carried on for 4 weeks.	DHEA had (*p* < 0.05) higher primordial (14.77 ± 0.6994 vs. 11.58 ± 1.264), primary (18.42 ± 0.7732 vs. 16.25 ± 0.5383) and secondary (8.583 ± 0.8744 vs. 8.333 ± 0.6195) follicle count vs. control. %TUNEL-positive Follicles lower in DHEA vs. control (40.77 ± 1.382 vs. 45.59 ± 1.653, *p* < 0.05) as well as %γ H2AX-positive follicles (53.77 ± 1.424 vs. 61.16 ± 2.288, *p* < 0.05)	DHEA promotes folliculogenesis, inhibits ovarian apoptosis and decreases DSB occurrences in oocytes
Ma et al., 2022 [91]	Hyperbaric oxygen treatment (HBOT)	Mouse study: 40-week-old female mice were administered 100% oxygen at 2.5 ATA pressure for 90 min. Duration: 10 consecutive days. Afterwards, mice were sacrificed for oocyte analysis and extraction for IVF	There were significantly higher serum AMH levels vs. control (~550 ng/L vs. ~250, *p* < 0.05). Significantly lower apoptotic nuclei vs. control (mean density ~55 vs. ~45, *p* < 0.05). The number of retrieved oocytes and number of blastocyst formation were significantly higher vs. control as well (7 vs. 3, *p* < 0.05 and 5 vs. 1, *p* < 0.05, respectively)	HBOT may improve the quality of oocytes
Xiao et al., 2022 [92]	Macrophage-derived extracellular vesicles	Mouse study: 10-month-old ICR female mice were injected with either PBS (control), M1 or M2 extracellular vesicles. Injections occurred five times over a period of 5 days	The M2 group had a significantly higher number of primordial cells (~45 vs. ~38, *p* < 0.05). Growing follicle % was significantly lower in the M2 group vs. control (~35% vs. ~25%). Relative intensity of p-AKT significantly was higher in M1 vs. control (1.5 vs. 1.25, *p* < 0.05) and significantly lower in M2 vs. control (1 vs. 1.25, *p* < 0.05). Intensity of p-RSP6 was significantly higher in M1 vs. control (1.2 vs. 1, *p* < 0.05) and significantly lower in M2 group (0.75 vs. 1, *p* < 0.05). Post-superovulation, the M2 group had a significantly higher number of retrieved oocytes vs. control (~12 vs. ~7, *p* < 0.05)	M2 extracellular vesicle injection may alleviate symptoms of ovarian aging through inflammation reduction
Liu et al., 2022 [93]	Cell-free fat extract (CEFFE)	Mouse study: Mice were divided into three groups: (A) 8-week-old mice (young control), (B) 10-month-old mice (old control) and (C) 10-month-old mice, given 200 μL CEFFE through the tail every other day. Study duration: 2 weeks	At 8 weeks post-treatment, the study group had significantly higher AMH levels compared to the control (~2300 pg/mL vs. ~2000, *p* < 0.001). It also had a significantly higher no. of primary (~4 vs. ~2, *p* < 0.05), secondary (~6 vs. ~2, *p* < 0.05) and total follicles (~14 vs. ~4, *p* < 0.01), as well as a significantly greater blastocyst formation rate (41.67 ± 37.27 vs. 3.57 ± 9.45, *p* < 0.05) and litter number (6.63 ± 2.07 vs. 3.25 ± 2.05 *p* < 0.01)and significantly lower relative γH2AX staining than the control (~5 vs. ~10, *p* < 0.01)	While the exact mechanism remains unknown, CEFFE provides and restores function in aged ovaries
Qin et al., 2022 [94]	TrkB agonist (Ab4B19)	Mouse study: Six 12-month-old C57BL female mice were divided into two groups, one treated with Ab4B19 and the other with just IgG. Duration: 16 days. The mice were then either mated to analyse fertility or were sacrificed for ovarian examination	There was a significantly (*p* < 0.05) higher primordial (~500 vs. ~400), preantral (~350 vs. ~250) and antral follicle count (~250 vs. ~150) per ovary vs. vehicle and significantly lower atretic follicle count vs. vehicle (~100 vs. ~150, *p* < 0.0001). There were significantly more ovulated oocytes per mouse (~28 vs. ~18, *p* = 0.0284) and a significantly larger mean litter size in the TrkB group vs. vehicle (~8 vs. ~6, *p* = 0.0017)	TrkB agonist treatment may have potential in prolonging the reproductive window as well as treating ovarian aging

## Data Availability

No new data were created.

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
