# Peer review of "Metabolic Mechanisms and Potential Therapeutic Targets for Prevention of Ovarian Aging: Data from Up-to-Date Experimental Studies"

_ijms, 2023, doi:10.3390/ijms24129828_

Round 1

Reviewer 1 Report

The manuscript, “Metabolic Mechanisms & Potential Therapeutic Targets for Prevention of Ovarian Aging: Data from Up-to-date Experimental Studies” by Valtetsiotis et al. examined potential novel metabolic mechanisms involved in ovarian aging according to recent data and how these mechanisms may be addressed through new potential medical treatments. Also it examined novel medical treatments currently available based mostly on experimental stem cell procedures as well as caloric restriction (CR), hyperbaric oxygen treatment and mitochondrial transfer.

Comments and Suggestions for Authors:

1- Please correct the keywords according to MeSH.

2- Put a comma between two numbers in line 27.

3- In line 145, correct the word "lin.es".

4- Where is section "4.1." in "4. Potential Molecular/Metabolic Treatments for Prevention of Ovarian Aging" part?

5- Add a table for section "4.2. Stem cells".

In this table, list the studies done by stem cells that are related to this field and bring the type of study, type of cells and other information in the table.

6- Why use the same numbers to classify "4.2.3 Human Placenta-Derived Mesenchymal Stem Cells (hPD-MSC)", "4.2.3 Menstrual Blood-derived Mesenchymal stem cells" and "4.2.3. Human endothelial progenitor cells" parts?

Please pay attention to the classification.

7-The section number "5.2.4. Adipose derived stem cell conditioned medium (ASC-CM)" should also be corrected.

8- In this article, there is no table that the reader can have a comprehensive overview of previous studies.

Please add several tables for better and easier understanding of previous studies.

9- Please add a conclusion section.

10- In this study, the use of 6 types of stem cells is mentioned. Have other stem cells not been used?

Reviewer 2 Report

The manuscript “Metabolic Mechanism & Potential Therapeutic Targets for Prevention of Ovarian Aging: Data from Up-to-date Experimental Studies” concludes current situations of possible mechanisms and potential therapeutic factors for the prevention of ovarian aging. Personally, I believe that the accumulation of such information would be contributed to clinical research for the prevention of ovarian aging in the future. If possible, it is necessary to consider how to solve the problem of ovarian aging prevention and approach it for application in clinical treatment. Also, the authors have already reported and trialed using their methods for the prevention of ovarian aging. Therefore, the authors should add some schemes based on their results to increase the reader’s understanding.

Some points have to be corrected.

Major points

1. In this review, the authors introduced new molecular factors involved in ovarian aging, such as DNA damage, cohesins, oxidative damage and so on. However, these factors are general factors for female infertility. I think the authors need to amend the subtitle.

2. The authors introduced the approach of using stem cells as a potential treatment for the prevention of ovarian aging in depth. If possible, I recommend adding some schemes based on their results to increase the reader’s understanding.

3. Main cause of ovarian aging is oxidative damage. So, It is better to add a schematic model of the relationship between each factor with oxidative damage. A schematic model can help to understand which treatment is more effective for decreasing oxidative damage.

Minor points

1. In line 10, please add a comma after “review”.

2. In line 172, please delete “by”.

3. In line 172, please amend “reactive oxygen species” to “ROS”.

4. In subtitle 4, where is 4.1.?

5. In line 577, please amend “signficantly” to “significantly”.

6. In line 642, please amend “reseachers” to “researchers”.

7. In line 644, amend “appllication” to “application” and “ooocytes” to “oocytes”.

Although this manuscript is well-summarized, it is inconvenienced by not having a figure and scheme model to understand this review’s contents. If the authors do not want to add some figures and scheme models, I think this review is barely suitable for the paper of IJMS. At that time, I recommend it is submitted to JPM or children, MDPI.

Round 2

Reviewer 1 Report

All of my suggestions are considered.

Reviewer 2 Report

I think that the revised manuscript has been fundamentally improved and that it includes the contents requested by the referees and editorial team. 

Totally, English language is fine. Please, check one more whether minor editing of English language required.